

# Development of an inexpensive matrix-assisted laser desorption—time of flight mass spectrometry method for the identification of endophytes and rhizobacteria cultured from the microbiome associated with maize

Michael G. LaMontagne[1], Phi L. Tran[1], Alexander Benavidez[2] and Lisa D. Morano[2]

[1] Department of Biology and Biotechnology, University of Houston, Clear Lake, Houston, Texas, United States
[2] Department of Natural Sciences, University of Houston, Downtown, Houston, Texas, United States

Corresponding author
Michael G. LaMontagne, lamontagne@uhcl.edu

## ABSTRACT

Many endophytes and rhizobacteria associated with plants support the growth and health of their hosts. The vast majority of these potentially beneficial bacteria have yet to be characterized, in part because of the cost of identifying bacterial isolates. Matrix-assisted laser desorption-time of flight (MALDI-TOF) has enabled culturomic studies of host-associated microbiomes but analysis of mass spectra generated from plant-associated bacteria requires optimization. In this study, we aligned mass spectra generated from endophytes and rhizobacteria isolated from heritage and sweet varieties of *Zea mays*. Multiple iterations of alignment attempts identified a set of parameters that sorted 114 isolates into 60 coherent MALDI-TOF taxonomic units (MTUs). These MTUs corresponded to strains with practically identical (>99%) 16S rRNA gene sequences. Mass spectra were used to train a machine learning algorithm that classified 100% of the isolates into 60 MTUs. These MTUs provided >70% coverage of aerobic, heterotrophic bacteria readily cultured with nutrient rich media from the maize microbiome and allowed prediction of the total diversity recoverable with that particular cultivation method. *Acidovorax* sp., *Pseudomonas* sp. and *Cellulosimicrobium* sp. dominated the library generated from the rhizoplane. Relative to the sweet variety, the heritage variety c ontained a high number of MTUs. The ability to detect these differences in libraries, suggests a rapid and inexpensive method of describing the diversity of bacteria cultured from the endosphere and rhizosphere of maize.

## INTRODUCTION

Current agricultural practices will not meet the nutritional needs of a population that will reach nine billion people by the middle of this century (*Anand, Grayston & Chanway, 2013*). In parallel, climate change will increase extreme weather events, including drought (*Dai, 2011*; *Trenberth et al., 2014*), while urbanization will reduce arable land (*Song, Pijanowski & Tayyebi, 2015*). Microbial products can mitigate this food crisis by increasing crop yield (*Tkacz & Poole, 2015*) and helping crops tolerate drought and marginal soils (*Brígido & Glick, 2015*). The plant microbiome abounds with endophytes and plant growth promoting rhizobacteria (PGPR) that can help plants recover more nutrients from the soil and tolerate stressors like drought (*Barnawal et al., 2013*; *Bresson et al., 2014*). PGPR can also control plant pathogens (*Chowdhury et al., 2013*), promote beneficial mycorrhizae colonization (*Labbé et al., 2014*) and produce potentially valuable secondary metabolites (*Brader et al., 2014*; *Kumar, Dubey & Maheshwari, 2012*; *Raaijmakers & Mazzola, 2012*); however, the vast majority of microbes associated with agrosystems, have yet-to-be cultured. This limits the development of microbial products from PGPR and manipulative experiments with defined and representative plant microbiomes that can facilitate hypothesis testing, as demonstrated with maize (*Niu et al., 2017*).

Given its long history of artificial selection and its importance as a global food crop, the maize microbiome is of particular interest. Maize stems (*Kämpfer et al., 2016a*, *2016b*) have yielded novel bacterial species and maize kernels host diverse bacterial endophytes (*Rijavec et al., 2007*). Few studies have compared the diversity of libraries of bacteria isolated from both the rhizosphere and endosphere of maize (*McInroy & Kloepper, 1995*). Further, although Bt and non-Bt modern maize showed little differences in their bacterial communities (*Mashiane et al., 2017*), maize varieties may support different microbiomes. For example, a comparison of libraries generated from a wild maize and a more modern variety yielded three PGPR species from the wild maize that were not recovered from the modern variety (*Mousa et al., 2015*) and ancient maize yielded a bacterial endophyte that inhibits multiple fungal pathogens (*Johnston-Monje & Raizada, 2011*). A deeper understanding of interactions between maize varieties and the microbiomes they host in their endosphere and rhizosphere could inform searches for PGPR but development of deep libraries of isolates for research and microbial product development multiplies the costs associated with microbial identification.

Matrix-assisted laser desorption-time of flight (MALDI-TOF) appears well suited for analysis of libraries isolated from plant microbiomes (*Ghyselinck et al., 2013*). MALDI-TOF systems provide strain-level identification of microbes (*Ahmad, Babalola & Tak, 2012*; *Sauer et al., 2008*; *Singhal et al., 2015*) for pennies an isolate. These systems compare favorably to the widely used method of 16S rRNA sequencing (*Emami et al., 2016*); however, the costs of acquiring and operating these systems and the challenges in interpreting the data they generate have limited the widespread application of MALDI-TOF. These systems use pattern matching between mass and reference spectra; however, these databases have poor representation of environmental microbes. This paucity of representation limits MALDI-TOF effectiveness (*Singhal et al., 2015*).

Compilations of spectra are available (*Böhme et al., 2012*; *Murugaiyan et al., 2018*; *Rau et al., 2016*). These compilations are typically species-specific databases of mass spectra, including databases for *Mycobacterium kansasii* (*Murugaiyan et al., 2018*) and *Vibrio* species (*Erler et al., 2015*) or for particular systems, like spacecraft (*Seuylemezian et al., 2018*). Further, custom databases require proprietary software and do not facilitate investigators sharing mass spectra, which would benefit science. To address this, several teams have developed applications for microbial identification by MALD-TOF by matching of spectra to peaks inferred from genomic (*Tomachewski et al., 2018*) or proteomic databases (*Cheng, Qiao & Horvatovich, 2018*) and cluster analysis allows for rapid dereplication without a reference library (*Clark et al., 2018*; *Dumolin et al., 2019*). Web applications for this analysis are available (*LaMontagne et al., 2017*); however, data analysis of mass spectra generated from bacteria isolated from the plant microbiome needs improvement (*Huschek & Witzel, 2019*). To our knowledge, a consensus designation of how clusters defined by MALDI-TOF corresponds to biological relevant taxonomic units has not emerged.

In this study, we generated a library of readily-culturable bacteria from the endosphere and rhizosphere of two varieties of maize (*Zea mays*): an agronomically important hybrid and a heritage variety. These maize models enabled comparisons of libraries that we predict would differ. Isolates were sorted, using custom scripts, into MALDI-TOF Taxonomic Units (MTUs) based on similarity of mass spectra generated by MALDI-TOF. These clusters, which are analogous to the operational taxonomic units widely reported in metagenomic studies, corresponded to a threshold of cosine similarity coefficient >0.65. These MTUs appeared coherent, as isolates clustered within an MTU showed nearly identical rRNA gene sequences and MTUs were consistent with species identified with a commercial mass spectra database. The throughput of this approach enabled comprehensive sampling of the aerobic, readily-culturable bacteria of the maize microbiomes we selected as a model. Rarefaction analysis indicated that the library we generate provided >70% coverage of the aerobic, readily-culturable bacteria in the maize microbiomes we sampled. This sampling depth facilitates testing hypotheses about what controls the diversity of libraries generated from plant microbiomes. For example, in this study, we observed that the rhizoplane of the heritage variety yielded the most MTUs that were unique to that particular niche compared to libraries generated from either the endosphere or the rhizoplane of the sweet variety.

## MATERIALS & METHODS

### Sampling maize microbiome

Two maize varieties were chosen for this experiment, a 'Heritage' and a 'Sweet' variety. The Heritage variety was Dent Earth Tones Corn (Botanical Interest Inc., Broomfield, CO, USA). This is a heritage variety selected for propagation based on its colorful kernels. The Sweet variety, Bodacious Hybrid Sweet Corn (American Seed Co, Spring Grove, PA, USA), has been highly bred for both sweetness and disease resistance. This modern variety has the designation "se" which stands for sugar enhancer and "R/M" indicating disease resistance to multiple maize diseases.

Plants were grown inside at the University of Houston-Downtown in fall of 2017. Two seeds per variety were planted in 3.5 inch × 3.5 inch pots containing LadyBug Vortex Potting Soil® (New Earth, Conroe, TX, USA). Plants were grown at room temperature in a two-tier plant growing unit with grow lights. The plants for this experiment were all on the same tier in two batches. Grow lights were set approximately 0.7 m above the plants and set for 12/12 h light/cycle. Pots were watered every 2–3 days. One batch, designated "New" was harvested 60 days after sowing. Another batch, designated "Old" was harvested 120 days after sowing. All plants were brought to the University of Houston—Clear Lake for plant measurements and microbial extraction.

Investigation of both maize varieties began with the cutting of plants into shoots and roots. Soil was knocked from the roots and the fresh weight of shoots and roots was recorded. The height of each shoot was recorded from the soil line to the longest shoot tip. Roots were transferred into a 50 ml conical tube and wetted with 40 ml of 5% DMSO prepared in sterile distilled water. The tubes were then vortexed for 1 min and centrifuged at $4,800 \times g$, in a swing-out rotor, for 10 min at room temperature. The roots were recovered with tweezers and blotted dry to obtain wet weights. The supernatant was discarded and the remaining slurry (~5 ml) was transferred to microtubes with a wide orifice pipet.

Endophytes were recovered from stem sections that were cut into lengths that weighed between 0.20 and 0.25 grams. These pieces were submersed in 70% ethanol for 60 s followed by flaming to remove ethanol. The pieces were then rinsed in sterile distilled water for 60 s. The pieces were then cut into approximately 1 mm fragments and ground to a slurry in 2 ml of sterile water, using a surface sterilized mortar and pestle (*Johnston-Monje et al., 2014*).

Slurries generated from roots and shoots were serially diluted in sterile phosphate buffered saline and inoculated on half strength tryptic soy broth (EMD Millipore, Billerica MA) solidified with 15 g per liter with agar (AmericanBio, Natick, MA) to make TSA plates. The track dilution method was used to spread dilutions (*Jett et al., 1997*). This method entailed spotting 10 µl from four dilutions in a row along one edge of the plate. The plate was then tilted to allow the droplets to run in a track across the plate. The plates were allowed to dry before being sealed with parafilm and incubated at 30 °C for 48 h. Colonies were picked from dilutions (generally the $10^{-4}$ dilution) that yielded isolated colonies. To maximize diversity, morphology was considered in picking colonies. Representative morphologies were selected from each track. Isolates were re-streaked to fresh plates twice for purity.

## Microbial Identification—MALDI-TOF

Isolates were prepared for MALDI-TOF analysis with the ethanol inactivation and formic acid extraction protocols recommended by Bruker Scientific (Billerica, MA, USA). This approach follows the recommendations of Freiwald and Sauer (*Freiwald & Sauer, 2009*), except 50 µl volumes of formic acid and acetonitrile were used. Briefly, bacteria were cultured overnight at 30 °C on TSA plates. Resulting colonies were then suspended in 300 µl of HPLC-grade water and treated by adding 900 µl HPLC-grade ethanol.

The resulting slurry was stored at 4 °C for one week. The ethanol treated cells were then recovered by centrifugation (16,000 × g, 2 min) and the resulting pellet was extracted with formic acid/acetonitrile to yield a formic acid extract.

Matrix solution, 10 mg/mL a-cyano-4-hydroxycinnamic acid (HCCA, Sigma, St. Louis, MO, USA) in 50% acetonitrile, 47.5% water, and 2.5% trifluoroacetic acid (ACN, Sigma, St. Louis, MO, USA) was prepared fresh for each use. Bacterial test standard (BTS, Bruker p/n 8255343) was dissolved in 50 µL of 50% aq. ACN, 2.5% TFA following the manufacturer instructions. MALDI target plate (MSP 96 polished steel target, Bruker p/n 8280800) was washed with trifluoroaceitic acid and 70% ethanol as recommended by Bruker. On one or two spots on the target, 1 µl of formic acid extract was spotted on the target followed by 1 µl of matrix solution. Two BTS spots were prepared by applying 1 µl of BTS solution followed by 1 µl of matrix solution to each spot. Two matrix blank spots were included on each target as negative controls. Targets were allowed to dry at room temperature for approximately 15 min and they were shipped overnight, with an ice pack, to the Proteomics and Mass Spectrometry Core Facility at the Huck Institute (The Pennsylvania State University, University Park, PA, USA). Positive-ion mass spectra were acquired on a Bruker Ultraflextreme MALDI TOF/TOF mass spectrometer. Linear detection mode with the following parameters was used: pulsed ion extraction 170 ns; Ion source 1 25 kV; Ion source 2 94% of Ion source 1, and Lens 32% of Ion source 1. Matrix suppression (deflection) was set to 1,500 m/z. The laser repetition rate was 667 Hz; Smartbeam parameter set to "3_medium". Real-time smoothing was Off, baseline offset 0.2%, analog offset 2.1 mV, and the detection was set for low mass range, 1,880–20,000 Da. The target was moved in a random walk, complete sample pattern; 50 shots were fired at 24 raster spots (1,200 total shots) limited to a 2,000-mm diameter.

Mass calibration was performed with BTS as the standard; quadratic calibration curve was constructed based on m/z values of 8 calibrants over 3,637 to 16,952 m/z range. Mass spectra were smoothed using 10 cycles of SavitzkyGolay, with width 2 m/z; and baseline-subtracted using TopHat algorithm. Mass List Find parameters were as follows: peak detection algorithm Centroid, S/N threshold 2, minimum intensity threshold 600, max number of peaks 300, peak width 4 m/z, height 90%. The processed mass spectra were loaded into the MALDI Biotyper Version 3.1 (build 66) software (Bruker) and searched against a Bruker Taxonomy library containing 7,854 entries.

Mass spectra were also analyzed by cluster analysis using an R script that implemented functions in MALDIquant (*Gibb & Strimmer, 2012*), PVclust (*Suzuki & Shimodaira, 2006*) and several other packages. This script included two optimization loops that iteratively sampled random values, within specified ranges, for seven parameters: half-window for smoothing, baseline removal, half-window for alignment, tolerance of alignment, signal to noise ratio (SNR) for alignment, half-window for peak detection and SNR for peak detection. The first loop identified the parameters that optimized the number of peaks shared, as Jaccard coefficients calculated with the philentropy (*Drost, 2018*) between pairs of average mass spectra generated from BTS. The Jaccard coefficients were selected to minimize the variability introduced by peak heights, as described previously (*AlMasoud et al., 2014*). Output from the first optimization was passed to a step then conducted

quality control analysis to identify noisy spectra. On the spectra that passed quality control, the second loop selected the parameters that minimized the overlap in cosine similarity values, calculated following (*Strejcek et al., 2018*), between closely related and distantly related isolates. The cosine similarity coefficient was selected because these similarities, which are analogous to Pearson coefficients, are the most widely used metric for assessing the similarity between mass spectra (*Huber et al., 2021*) and are readily calculated with the R package coop (*Schmidt, 2019*). Cosine similarities were weighted by Jaccard coefficients normalized after with the formula $y = y_0 + x/(x + 0.2)$, where x is Jaccard coefficient, $y_0$ is the average cosine similarity when $x = 0$ and y is a predicted cosine value. The script then performed cluster analysis to define MALDI-TOF taxonomic units (MTUs), rarefaction analysis using the iNext package (*Hsieh, Ma & Chao, 2016*) and trained a machine learning algorithm using the RWeka package (*Hornik, Buchta & Zeileis, 2009*). This script is presented in one R markdown file (Supplemental Data). Raw mass spectra and Bruker system identifications are available as dataset MSV000086274 in MassIVE (*Deutsch et al., 2016*).

Curve fitting and ANOVA testing of significance of correlations were completed with SigmaPlot for Windows version 13.0 (Systat Software, San Jose, CA, USA).

## Microbial Identification—16S rRNA gene sequencing

Representative isolates of MTUs that were not identified at the species level using the Biotyper database and software (Biotyper score < 2), were identified by 16S rRNA gene sequencing. DNA was extracted from isolates cultured overnight in tryptic soy broth (30 °C, 200 rpm) with the Puregene kit, following the manufacturer's protocol (Qiagen, Carlsbad CA, USA). A near intact fragment of the 16S rRNA gene was amplified with ReadyMade™ Primers 16SrRNA For and 16srRNA Rev supplied by IDT (Coralville, IA, USA) using DreamTaq Hot Start DNA polymerase, following the manufacturer's protocol (ThermoFisher, Waltham, MA, USA). PCR reactions were preheated (94 °C, 2 m) and then cycled 32 times through the following steps: denaturing (94 °C, 30 s), annealing (56 °C, 30 s) and extension (72 °C, 90 s). The final extension step was extended for 5 min and product size was confirmed by electrophoresis with a FlashGel™ DNA Kit (Lonza, Basel, Switzerland). PCR products were then purified with the DNA Clean & Concentrator kit, following the manufacturer's protocol (Zymo, Irvine, CA, USA). Purified fragments were sequenced with the above ReadyMade™ Primers, using Sanger technology, by Lone Star Laboratory (Houston, TX, USA).

Single-pass, 16S rRNA sequences were manually curated in BioEdit (*Hall, 1999*), checked for chimers with Decipher (*Wright, Yilmaz & Noguera, 2012*) and aligned against curated sequences with SINA (*Pruesse, Peplies & Glöckner, 2012*). ModelFinder (*Kalyaanamoorthy et al., 2017*) was used within the IQ-TREE environment (*Trifinopoulos et al., 2016*) to select the appropriate phylogenetic maximum likelihood model. Bootstrap values were calculated with UFBoot2 (*Hoang et al., 2017*) and a tree was generated with Newick Display 1.6 (*Junier & Zdobnov, 2010*). Decipher, ModelFinder, UFBoot2 and Newick Display were all run with default parameters. The 16S sequences were uploaded to GenBank and assigned accession numbers MW092913–MW092939.

Table 1 Number of maize-associated isolates analyzed by MALDI-TOF.

| | Maize Variety | |
| Niche | Heritage | Sweet |
|---|---|---|
| Endosphere | 22 | 23 |
| Rhizoplane | 51 | 36 |

# RESULTS

The majority of bacteria isolated from maize did not match reference spectra in the Bruker system. Of mass spectra generated from the 132 representative colony morphologies cultured from the rhizoplane and endosphere of two varieties of maize (Table 1), ten failed the initial quality control of the Bruker system and eight failed a SNR threshold we describe below. Of the remaining 114 isolates, 36 were identified at the species level with probable confidence, as defined by a classification score of > 2 with the Biotyper system. Mass spectra from these 114 isolates were clustered into 60 MTUs with the following pipeline. These MTUs agreed with operational taxonomic units assigned by 16S rRNA gene sequencing.

Iteratively trying different parameters, like tolerance and SNR for aligning mass spectra and detecting peaks respectively, identified a set that optimized the number of peaks shared between spectra generated from BTS references spotted on separate targets. The number of peaks shared between reference spectra reached an asymptote after a dozen iterations (Fig. S1). Extending the number of iterations of the optimization loop to 2,000 identified parameters that corresponded to the highest Jaccard similarity coefficients (0.952), in terms of peaks matched between spectra generated from BTS. Pairwise comparisons of these BTS spectra shared 20/21 peaks and showed cosine similarities that ranged from 0.963–0.970. Jaccard similarity coefficients between these reference spectra showed a modal relationship with the total number of peaks detected in this set of samples (Fig. S2 top). The zenith of this relationship occurred between 389–406 total peaks detected from the 174 spectra that passed initial quality control within the Bruker system. The parameters that yielded the most reproducible peaks were selected for identification of quality spectra, as defined by the number of the SNR for the ten largest peaks and the number of peaks detected. Quality control analysis, using a threshold of a median SNR of 15 for the 10 largest peaks and detection of at least 13 peaks, pruned mass spectra generated from 8 of 122 isolates from subsequent cluster analysis. The second optimization loop found, for pairs of isolates that had practically identical 16S rRNA sequences (>99%), the average Jaccard coefficient for comparisons reached a zenith at 498 peaks detected from 114 isolates (Fig S2 bottom).

Cosine similarities calculated with the parameters that showed the greatest discrimination between comparisons within species than between species, suggested a hyperbolic relationship with the Jaccard similarities (Fig. 1). Most (6472/6670) of the pairwise comparisons showed Jaccard similarity coefficients of less than 0.2. In that range, cosine similarities averaged (±SD) 0.27 ± 0.10. For the 192 pairwise comparisons that

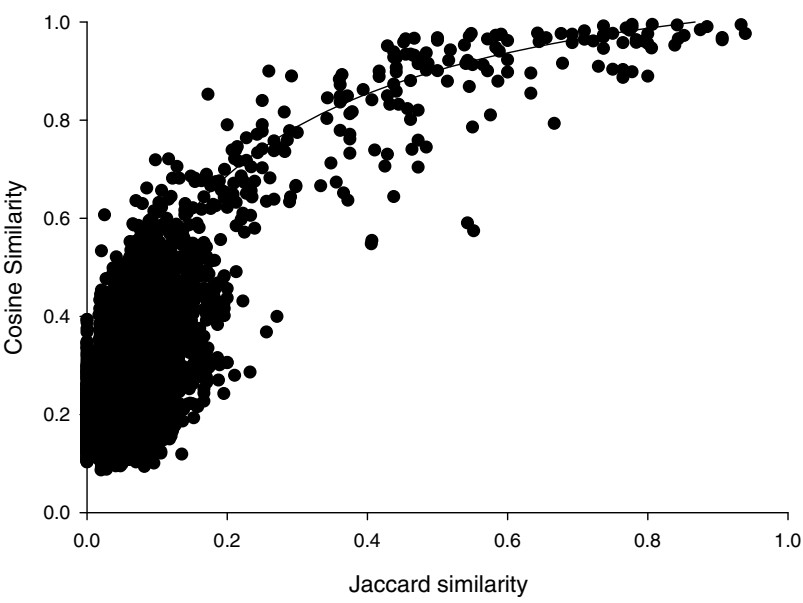

**Figure 1 Cosine versus Jaccard similarities for pairwise comparisons.** Line illustrates a mathematical model of the hyperbolic relationship described in Methods, not a fit to the data.

showed Jaccard similarities greater than or equal to 0.2, cosine similarities averaged 0.81 ± 0.16. Jaccard similarities of 0.2 corresponded to cosine similarities of about 0.69. This suggests that spectra that share very few peaks may show relatively high cosine similarities, which may reflect the variability in peak heights. Accordingly, we weighted cosine similarities by Jaccard coefficients (see "Methods").

The percent identities of pairwise comparisons 16S rRNA genes fit a sigmoidal model with weighted cosine similarities (Fig. 2). The fit had a modest correlation coefficient ($r^2 = 0.445$) and analysis of variance suggested the trend was highly probable ($P < 0.0001$); however, most (259/325) of the pairwise comparisons showed percent identities of 16S rRNA genes of less than 90%. The midpoint of this sigmoidal model corresponded to a cosine similarity of 0.66. Cosine similarities of >0.66 consistently corresponded to a percent identify of ribosomal sequences ranging from 99–100% (Fig. 2).

Inspection of Fig. 2 suggested that cosine similarity of 0.65 would distinguish species. Using that threshold, clustering of a library of 114 isolates into MALDI-TOF taxonomic units (MTUs) defined 60 MTUs. Of these, 25 clusters contained more than one isolate (replicated MTUs) and 35 were singletons. Extrapolation from rarefaction analysis predicted that a library of 228 isolates would contain 84 MTUs, with a confidence interval of 70–98 (Fig. 3 top). This library of isolates corresponds to 70% coverage, with a confidence interval of 62–77% (Fig. 3 bottom). In other words, as defined by MTUs, the pooled library contained the majority of bacterial species that could be readily isolated from the rhizoplane and endosphere of both maize varieties we sampled. Further extrapolation predicted that a library of 228 isolates would provide 78–95% coverage.

Partial sequencing of 16S rRNA genes yielded sequences that were used to classify isolates that were not matched to the commercial database into 18 phylotypes that
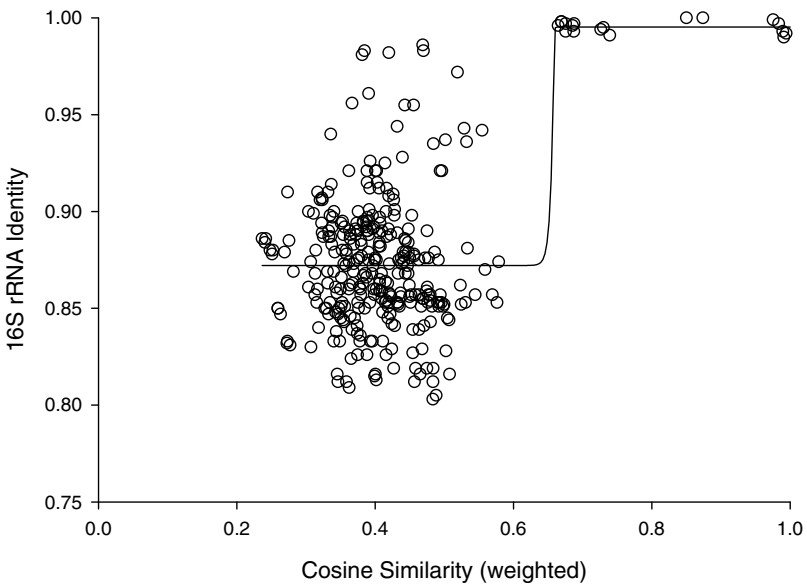

**Figure 2 Cosine similarity versus 16S identity.** Cosine similarity was weighted with values predicted from Jaccard coefficients using the hyperbolic model in Fig. 1 as described in Methods. Line represents a sigmoidal curve fit.                       

corresponded to different genera (Fig. 4). Phylotypes defined by rRNA gene sequencing generally appeared robust, as assessed by bootstrap values > 90% (Fig. 4). The exception was separation of the branch containing isolates that classified as *Paenibacillus* and *Bacillus* species, which bootstrapping supported only 76% of iterations. Within this Bacillales order, and in terms of rRNA gene sequences, isolate H1OZ89 showed high similarity to a phosphate solubilizing *Paenibacillus* species previously isolated from rape (*Ding et al., 2019*). Also, within this order, isolate H3OZ122 showed high similarity to a microplastics degrading *Bacillus* strain recently isolated from the Yellow Sea (*Wang et al., 2019*), isolate S4OZ125 showed high similarity to *Bacillus aciditolerans*, a recently described novel species isolated from a rice field (*Ding et al., 2019*) and H3OZ107 showed similarity to a *Staphylococcus hominis* strain isolated from a blood sample (Fig. 4). These isolates (H1OZ89, H3OZ122, S4OZ125 and H3OZ107) clustered into separate MTUs, which suggests the similarity threshold defined MTUs corresponds to species that occupy different niches.

Consistent with MTUs that classified within the Bacillales order, the topology of Fig. 4 generally showed congruence with MTUs. For example, MTUs 33, 34, 46 and 51 each contained two isolates that both clustered together in the same leaf on the tree. For the seven isolates of the *Acidovorax* genera shown in Fig. 4, the threshold for clustering appeared conservative. The rRNA sequences generated from seven isolates showed > 99% identity with species of *Acidovorax*, and each other, but were separated into three MTUs (29, 35 and 54). MTU 35 corresponded to a branch in Fig. 4 supported with 94% bootstrapping value, which suggests that cluster is coherent. On the other hand, MTUs 25 and 54 appear to have isolates of the same species, split into two clusters.

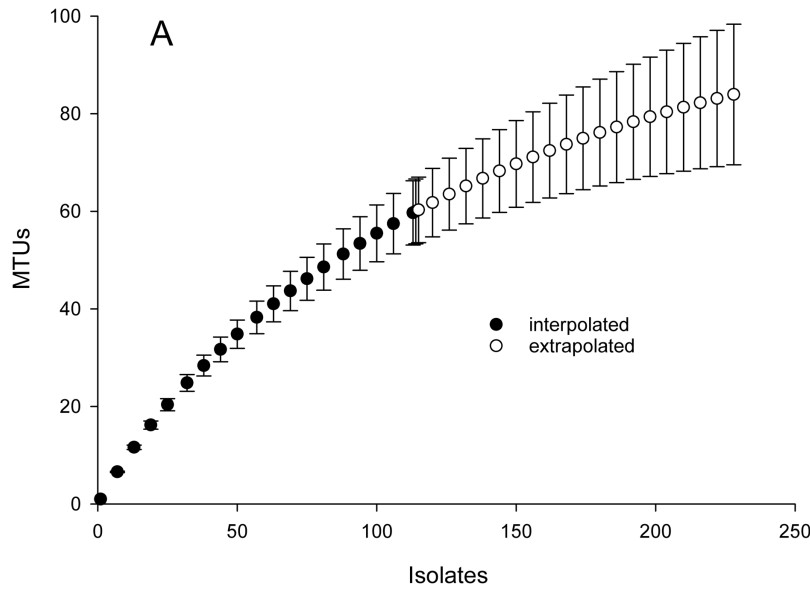

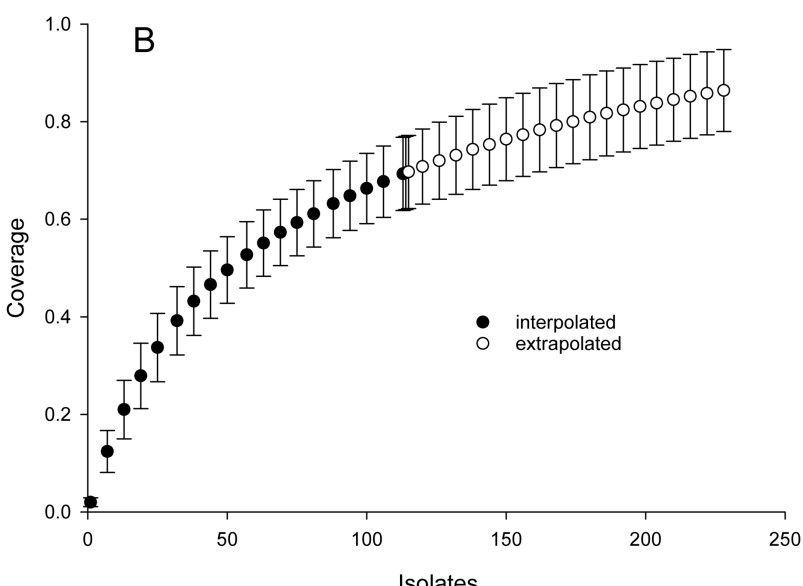

**Figure 3 Rarefaction analysis of library generated from the maize microbiome.** X-axis presents number of isolates in the library (filled circles) or an extrapolation from the existing library (open circles). Confidence intervals present standard error bars. Figures were generated with the default parameters in iNext. (A) Y-axis presents number of taxonomic units defined by similarity of mass spectra. (B) Y-axis presents fraction of total diversity recovered.

Matching to reference spectra did not appear to depend on the quality of the spectra, as classification scores showed no relationship with the SNR of spectra. The set of identified species contained 35 isolates that classified as bacteria and one isolate that classified as a fungal species (*Trichosporon mucoides*). Of the 35 bacterial isolates 6 were singletons. The 29 replicated isolates were classified into 11 species. These isolates generally corresponded to specific MTUs. For example, the 6 isolates reliably identified as

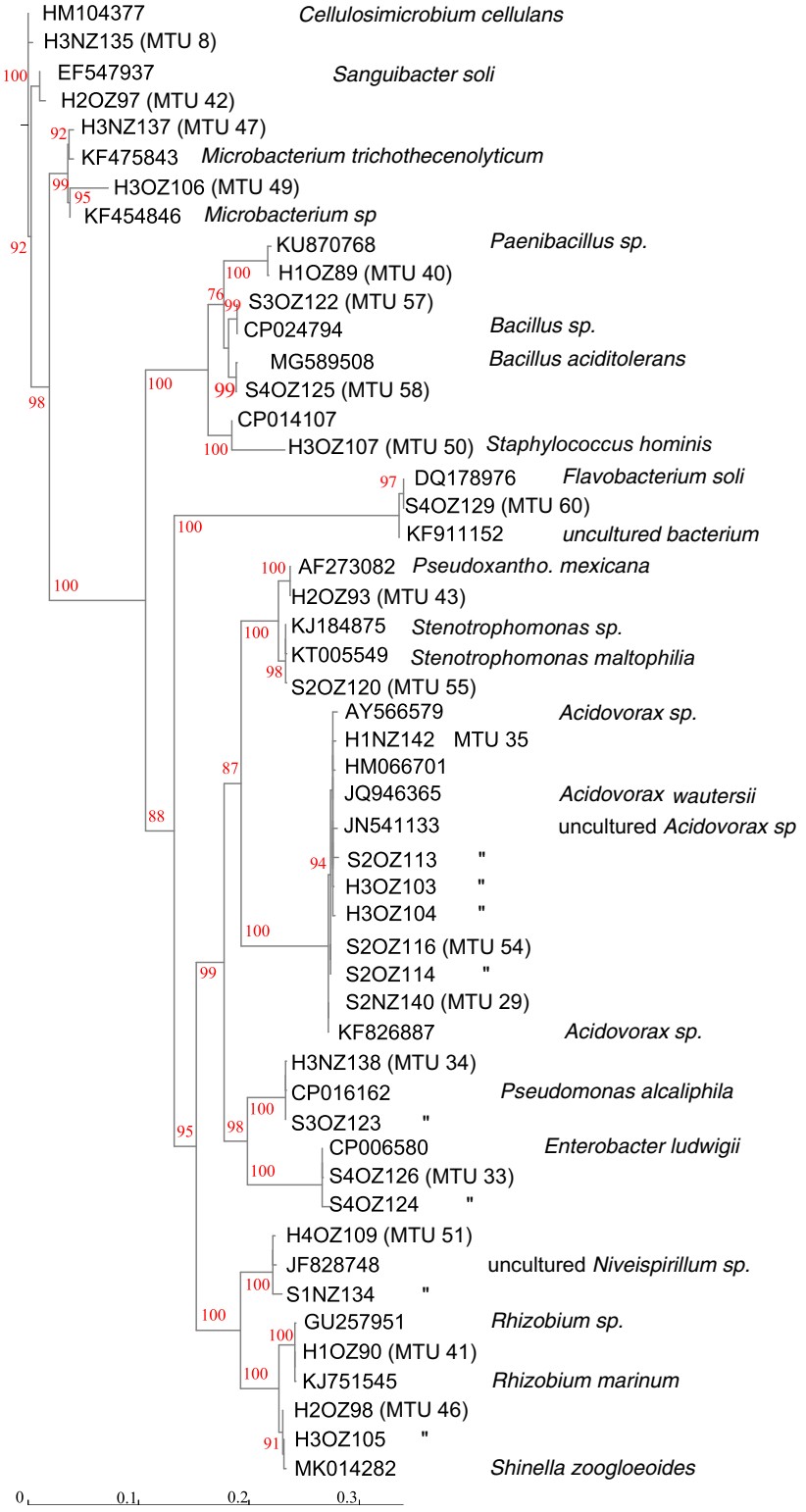

**Figure 4 Phenogram generated from 16S rRNA gene sequences.** Partial sequences were aligned against curated sequences with SINA. ModelFinder was used select the appropriate phylogenetic maximum likelihood model within IQ-TREE. Bootstrap values were calculated with UFBoot2. Genbank accession numbers and species identifications are provided for reference sequences. Scale indicates substitutions, where a distance of 0.04 corresponds to 96% identity.



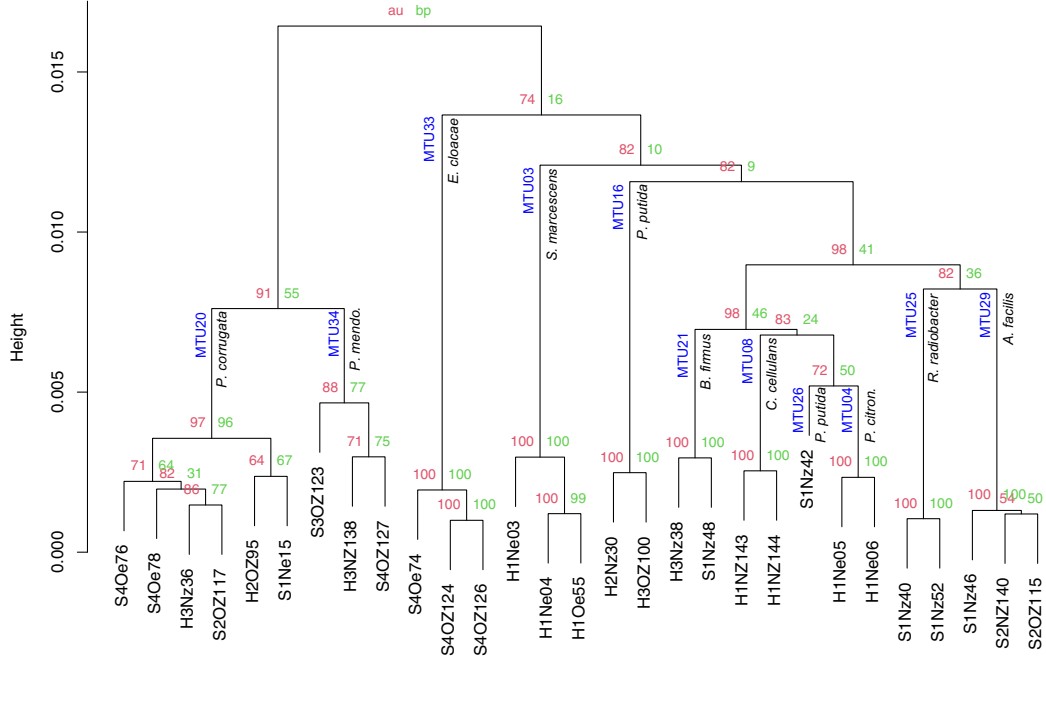

**Figure 5 Hierarchical clustering of mass spectra for isolates reliably identified with Bruker MALDI Biotyper system.** AU and BP values present approximately unbiased probability values and bootstrap probabilities assigned with pvclust. MTUs and species identification given by the Biotyper system are indicated at each node. Height presents dissimilarity between nodes.

*Pseudomonas corrugata* all clustered into MTU 20 (Fig. 5). Similarly, isolates identified by the Bruker system as *Pseudomonas mendocina, Enterobacter cloacae, Serratia marcescens, Bacillus firmus, Cellulosimicrobium cellulans, Pseudomonas citronellolis, Rhizobium radiobacter* and *Acidovorax facilis* clustered within MTUs 34, 33, 3, 21, 8, 4, 25 and 29 respectively (Fig. 5). Bootstrap values, for iterations of cluster analysis, generally supported the congruence of these MTUs with species identifications. Bootstrap values for these clusters exceeded 95%, with the exception of MTU 34 (Fig. 5). However, the dendrogram was not coherent with respect to the phylogeny of the isolates. In particular, *Pseudomonas* species were dispersed throughout the tree and appeared within a branch that contained Gram positive species (Fig. 5) and, for the three isolates the system identified as *P. putida*, the threshold used to define MTUs appeared conservative. These isolates were split into two MTUs (Fig. 5). The lone isolate in MTU 26 showed little similarity, in terms of cosine similarity, to any other isolate.

Machine learning resolved the apparent discrepancy for the isolates that clustered into different MTUs but were identified as one species by the Bruker system. To train the machine learning algorithm, MTUs were identified by the Bruker system and 16S

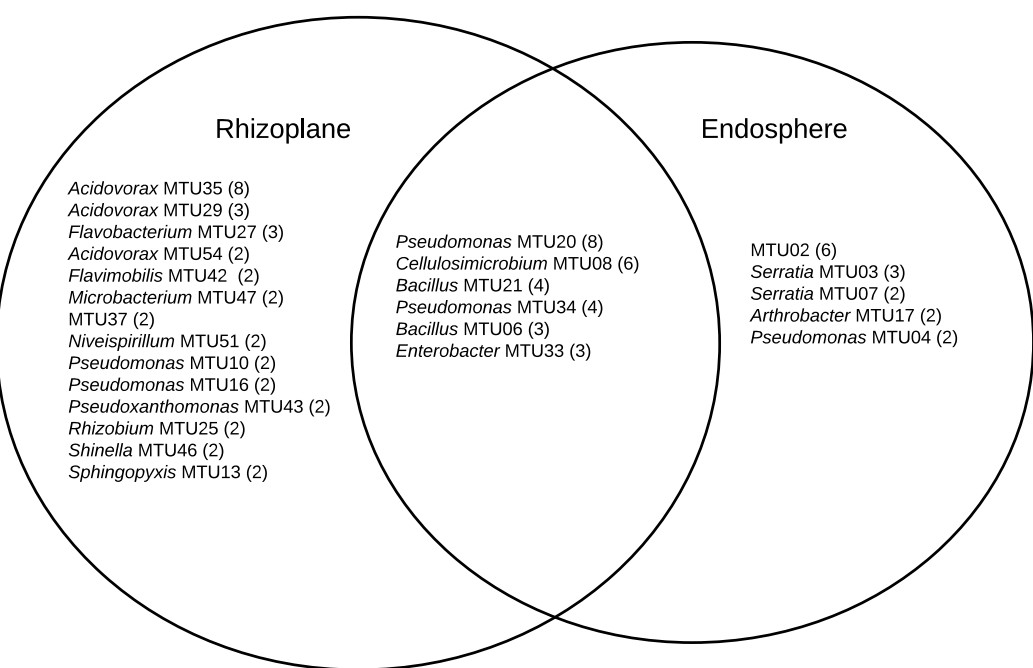

**Figure 6 MTUs detected in library generated from the rhizoplane and endosphere of maize.** The number of isolates clustered in each MTU is indicated. Singletons are not shown.

sequencing. When the Bruker and 16S identification differed, the 16S identification was used but this was rare. Identification by the Bruker system and 16S sequencing agreed at the genera level, for Bruker scores greater than or equal to 1.6; however, the two approaches showed some discordance at the species level. For example, MTU 33 was reliably identified (Bruker score = 2.5) as *Enterobacter cloacae* but these isolates clustered with *Enterobacter ludwigii* in Fig. 4. MTU 34 was probably identified (Bruker > 2) as *Pseduomonas mendocina* but these isolates clustered with *P. alcaliphila* in Fig. 4. MTU 35 were identified by the Bruker system as *Acidovorax facilis* with scores > 2.2 but a representative of that cluster showed high similarity, in terms of 16S sequence, to *A. wautersii* (Fig. 4). This MTU was defined as *A. wautersii*. When species were split into two MTUs, they were pooled. For example, the three isolates identified, at the probable level, as *P. putida* by the Bruker system were defined as that species, even though they belonged to two MTUs (Fig. 5). MTUs 54 and 29 were identified as one clade *Acidovorax sp.* based on Fig. 4. After discarding 34 singletons, this classification scheme provided at least genera level identities for all but two of the remaining isolates. The machine learning algorithm classified 100% of the 80 replicated isolates, and the two BTS references, correctly as 22 MTUs. This accuracy of classification is reflected in a Cohen's *Kappa* coefficient of 1, where this coefficient ranges from −1 to 1.

A combination 16S sequencing and MALDI TOF methods identified distinct MTUs that were relatively abundant to particular niches or plant varieties. Libraries generated from the rhizoplane contained the highest number of replicated MTUs overall (Fig. 6). Of those 20 MTUs, 14 did not appear in libraries generated from the endosphere. These

rhizoplane-specific MTUs included numerous representatives of two *Acidovorax* sp. and *Flavobacterium sp. Niveispirillum sp., Pseudoxanthomonas sp., Rhizobium radiobacter and Shinella sp.* (Fig. 6). MTUs specific to the endosphere included a yet-to-be identified cluster (MTU02), *Serratia* sp., *Arthrobacter* sp., and *Pseudomonas* sp. (Fig. 6). MTUs identified as *Pseudomonas* sp., *Cellulosimicrobium* sp.*, Bacillus* sp., and *Enterobacter* sp. were found in both the rhizoplane and the endosphere (Fig. 6). Libraries generated from the heritage variety contained more MTUs (11) unique to that variety than libraries generated from the sweet variety (Fig. S3).

## DISCUSSION

Optimization of cluster analysis of mass spectra generated from a library of bacteria isolated from maize microbiomes produced coherent MTUs. These MTUs agreed with species defined by 16S rRNA gene sequencing and by the Bruker system and with niches typically occupied by species related to these MTUs. The library of readily culturable bacteria isolated from the rhizoplane of the heritage variety contained the most strains that were specific to a particular niche. That is strains that abound in libraries generated from that sample and were not common in libraries generated from the endosphere of that variety or from rhizoplane and endosphere of the sweet variety. The rhizoplane and endosphere have many species in common, but it is logical that the rhizoplane community is more diverse given its direct contact with soil, which are diverse systems (*Howe et al., 2014*). In fact, endophyte communities may be a subset of the more extensive rhizosphere communities (*Long et al., 2010*).

The relatively high abundance of *Acidovorax* and *Flavobacteria* in libraries generated from the rhizoplane (Fig. 6) is consistent with previous culture-dependent surveys of plant microbiomes. *Acidovorax* is commonly found in both soil and plants (*Long et al., 2010*) and *Flavobacteria* genera includes a terrestrial clade associated with plant roots plant roots (*Kolton et al., 2013*). However, these genera are not consistently associated with maize roots, as assessed by metagenomic analysis (*Niu et al., 2017*). We also isolated *Rhizobium radiobacter (*formerly *Agrobacterium tumefaciens).* This soil bacteria interacts with plants and strains of this species show potential for bioremediation applications (*Deepika et al., 2016*) and novel species of this genera have been isolated from maize roots (*Gao et al., 2017*). *Microbacterium* has been isolated specifically from maize kernels (*Rijavec et al., 2007*). This endophyte may have been in the seed when it was sown.

The library generated from the maize endosphere included species that colonize plants. The most abundant MTU specific to the endosphere (MTU02) was not identifiable with the Bruker system and was not archived for future analysis. Other MTUs that abounded in libraries generated from the endosphere included *Serratia*, *Arthrobacter* and *Pseudomonas* species. *Serratia marcescens* is a common endophyte in cotton roots (*McInroy & Kloepper, 1995*) and rice (*Tan et al., 2001*). Several *Serratia* species associate with maize roots (*Mosimann et al., 2017*) and *S. marcescens* can cause maize whorl rot (*Wang et al., 2015*). *Arthrobacter spp.* are readily isolated from cotton stems (*McInroy & Kloepper, 1995*) and identified as an endophyte in maize sap (*Ali et al., 2018*).

MTUs found in both the libraries generated from the rhizoplane and endosphere included genera frequently associated with plant hosts. *Pseudomonas* sp. are readily cultured from the rhizoplane of corn on selective media (*Mosimann et al., 2017*). *Cellulosimicrobium sp.* can stimulate plant growth and act as a biocontrol for barley (*Nabti et al., 2014*). Recently the genome of a *Cellulosimicrobium* sp. strain isolated from endosphere of a perennial grass was sequenced because of its growth promoting properties (*Eida et al., 2020*). *Bacillus* sp. are ubiquitous in soils and widely used as microbial treatments of seeds (*Rocha et al., 2019*) and *Enterobacter* sp. can promote the growth of maize (*Naveed et al., 2014*) and appear a core member of that host's microbiome (*Niu et al., 2017*).

Many of the potentially beneficial rhizobacteria and endophytes presented above appear associated with only the heritage and not the sweet variety. For example, *Cellulosimicrobium* and *Serratia* sp. was common to only the heritage variety and *Enterobacter* and *Flavobacteria* sp. were common to only the sweet variety. This can inform microbial discovery efforts, where the goal is to isolate strains of a particular genera; however, for development of microbial products to improve the productivity and sustainability of maize, the variety cultivated by growers should be considered. Plant varieties form close associations with particular beneficial rhizobacterial strains (*Batstone et al., 2020*).

Clustering rhizobacteria and endophytes into MTUs, with the scripts presented herein, suggests a method to identify beneficial strains of plant-associated bacteria. MALDI-TOF has the resolution to differentiate strains of species that differ in functions, like antibiotic production (*Clark et al., 2018*) and resistance (*Flores-Treviño et al., 2019*). This resolution is particularly valuable in distinguishing *Bacillus* species (*Hotta et al., 2011*; *Lasch et al., 2009*). For example, MALDI-TOF can identify pathogenic strains of clinical (*Celandroni et al., 2016*) and environmental isolates of *Bacillus* (*Starostin et al., 2015*). In vitro assays, like 1-aminocyclopropane-1-carboxylate deaminase production (*Glick, 2014*), are widely used to identify potential PGPR. The correspondence between these phenotypes and MTUs requires further investigation.

Identification of maize-associated bacteria involved leveraging species identification with the Bruker system with 16S rRNA gene sequencing. This hybrid approach is widely used in applications of MALDI-TOF to environmental microbiology, where many of the isolates will not match the commercial database. For this study, the percentage of bacteria identified to the probable species level (32%) with the Biotyper system was lower than previous reports (73%) for soil bacteria (*Strejcek et al., 2018*) but in the range reported for a library generated from the rhizosphere of horseradish (*Uhlik et al., 2011*) and bacteria isolated from seawater (*Timperio et al., 2017*). We also observed discrepancies between species identifications for the two approaches, as reported previously and confirmed recently (*Pandey et al., 2019*). This could reflect the paucity of environmental isolates in the reference library we queried. The confidence of identification shows a hyperbolic relationship with the number of reference spectra in the database and approaches an asymptote at about 50 main spectra for each strain (*Erler et al., 2015*). The commercial database we queried only contains a few main spectra for most species.

With cluster analysis, we can begin to understand the diversity of rhizobacteria cultured with any particular technique from a plant's microbiome. Clustering of mass spectra, following optimization of alignment, identified MTUs that corresponded to highly similar (>99%) 16S rRNA gene sequences and species defined by a commercial database. The cosine similarity (0.65) selected as a threshold for defining MTUs was lower than threshold (0.79) reported by Strejcek et al. (*Strejcek et al., 2018*) but the coherence of the clusters, in terms of the consistency with clusters by 16S rRNA sequencing defined by a commercial spectra database and, supports this lower threshold. A more conservative threshold would split clusters that cannot be resolved with 16S rRNA sequencing, which is the current standard for microbial identification (*Edgar, 2018*).

High bootstrap support for a dendogram generated from mass spectra (Fig. 5) suggests the data analysis pipeline used to generated these clusters is robust. This suggests an approach to comprehensively compare libraries of bacteria isolated from different parts of the maize microbiome and test hypotheses about the efficacy of advanced culturing techniques, such as in situ cultivation (*Berdy et al., 2017*), improved media formulations (*Tanaka et al., 2014*) and gnotobiotic systems (*LaMontagne, 2020*). Machine learning also supported the clusters defined herein, and this could lead to an alternative method of identification (*De Bruyne et al., 2011*). Indeed, dozens of machine learning algorithms have recently been applied to this task (*Weis, Jutzeler & Borgwardt, 2020*). Our analysis suggests Rweka shows promise; however, development of such an identification system would require a challenge dataset generated externally, preferably on a different instrument (*Clark, Murphy & Sanchez, 2020*).

The reliance on 16S sequences limits the scope of this study. Phylogenetic analysis of the 16S rRNA gene benefits from a vast public database and user-friendly data analysis packages; however, analysis of this one gene lacks resolution. For example, diverse Bacillus species share identical rRNA gene sequences. Multilocus sequence typing (MLST) provides a higher resolution approach (*Maiden et al., 1998*) but MLST can fail to accurately estimate phylogenetic relationships between bacteria (*Tsang et al., 2017*). Whole genome sequencing provides strain-level identification (*Salipante et al., 2015*) but costs hundreds of dollars per isolate. Continued decreasing costs per bp of next generation sequencing (*Park & Kim, 2016*) and application of low-cost, third generation sequencers (*Petersen et al., 2019*) suggests whole genome sequencing will soon be standard for clinical applications (*Anis et al., 2018*); however, the true cost of whole genome includes library preparation, skilled labor and computational costs (*Sboner et al., 2011*). These costs limit the application of this approach to high-throughput microbial discovery programs.

The pooling of isolates collected at different stages of growth, and isolated on different media, into one library limits the scope of this study and precludes testing of hypothesis about the controls of the diversity and limits our ability to estimate the coverage these libraries provided. Defining the diversity of maize microbiome, within different locations of the host and between different varieties of maize, would require generation of much larger libraries then is feasible with culture-dependent techniques. Metagenomic analysis is

better suited testing hypotheses and provides a more complete census of the microbial communities in nature (*Handelsman, 2004*); however, this dataset revealed trends consistent with metagenomic studies, ecological theory and published observations.

In this study, we used the maize microbiome as model and demonstrate that MALDI-TOF systems provide a relatively low-cost method of identifying bacteria isolated from plant microbiomes. This proteomics approach is available to the general scientific community through core facilities, like the Huck Institute a Pennsylvania State University, which processed the samples herein. The costs of analysis by vendors is about a dollar per isolate for pre-spotted targets. The ethanol treatment/formic acid extraction applied herein is robust. Undergraduates in a teaching laboratory at the University of Houston—Clear Lake generated the protein extracts presented herein and dozens of students in this laboratory routinely generate quality spectra the first time they try this technique. The primary cost associated with this approach is purchasing of the reusable targets (~$500 each) and the scripts presented herein allow users to process the data with packages freely available in R.

The availability of MALDI-TOF analysis, through core facilities and commercial vendors, as well as the development of data analysis pipelines, like the one presented herein, should allow investigators to test the coherence of MTUs in terms of phenotypic traits and genotypes. However, to fully realize the potential of MALDI-TOF for identifying bacteria associated with plants, we need to build public databases of both raw and processed mass spectra and the metadata associated with the isolates. In other words, the community needs a resource analogous to the short archive available for next and third generation sequencing data. MassIVE provides such a resource (*Deutsch et al., 2016*) but only a handful of mass spectra of bacteria isolated from the plant microbiome have been archived in that database to date. We echo the call for the development and use of such a central depository of mass spectra generated from plant-associated microbes (*Ahmad, Babalola & Tak, 2012*) and environmental isolates in general.

## CONCLUSIONS

MALDI-TOF facilitated the classification of bacteria isolated from the maize microbiome into 60 MTUs. Our method proved to be a rapid and inexpensive method of describing the types of bacteria cultured from plant microbiomes with freely available spectra alignment and machine learning packages.

## ACKNOWLEDGEMENTS

The authors thank Dr. Tatiana Laremore at the Proteomics and Mass Spectrometry Core Facility (Pennsylvania State University) for MALDI-TOF mass spectra acquisition, MBT microorganism identification, and help with the manuscript preparation. The authors thank Torri Fugate-Mullins (UHCL) for help with microbial isolation and manuscript preparation. The authors thank two anonymous reviewers for suggesting revisions to an earlier version of this manuscript.

### Funding

This work was supported by student fees collected in teaching laboratories at UHCL and from the Faculty Research Support Fund at UHCL (No. A09S19). LaMontagne received support during writing of this manuscript from National Science Foundation (No. 2028400). Phi Tran received support from the grant Pathways to STEM Careers (No. P031C160242), funded by the HSI STEM program of US Department of Energy. There was no additional external funding received for this study. The funders had no role in study design, data collection and analysis, decision to publish, or preparation of the manuscript.

### Grant Disclosures

The following grant information was disclosed by the authors:
Faculty Research Support Fund at UHCL: A09S19.
National Science Foundation: 2028400.
Pathways to STEM Careers: P031C160242.
HSI STEM program of US Department of Energy.

### Competing Interests

Michael G. LaMontagne is an Academic Editor for PeerJ and the co-Founder of EndoBiome - a company that develops microbial products for agriculture.

### Author Contributions

- Michael G. LaMontagne conceived and designed the experiments, performed the experiments, analyzed the data, prepared figures and/or tables, authored or reviewed drafts of the paper, and approved the final draft.
- Phi L. Tran performed the experiments, authored or reviewed drafts of the paper, and approved the final draft.
- Alexander Benavidez performed the experiments, prepared figures and/or tables, and approved the final draft.
- Lisa D. Morano conceived and designed the experiments, performed the experiments, analyzed the data, authored or reviewed drafts of the paper, and approved the final draft.

### Patent Disclosures

The following patent dependencies were disclosed by the authors:
 LaMontagne MG. 2020. Gnotobiotic rhizobacterial isolation plant systems and methods of use thereof. US Patent 10801079, Oct. 13, 2020.

### Data Availability

 The 16S sequences are available at GenBank: MW092913 to MW092939.
 Raw mass spectra and Bruker system identifications are available in MassIVE: MSV000086274.

The Supplemental Files are comprised of the data analysis scripts in an R markdown file and in html format; the metadata and the 16S identity matrix inputs to this data analysis pipeline; and for convenience, R files are available as input files that allow investigators to input the results of multiple, computationally-intensive, iterations of alignment attempts.

## Supplemental Information

Supplemental information for this article can be found online at http://dx.doi.org/10.7717/peerj.11359#supplemental-information.

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
