# Peer review of "Development of an inexpensive matrix-assisted laser desorption—time of flight mass spectrometry method for the identification of endophytes and rhizobacteria cultured from the microbiome associated with maize"

_PeerJ, doi:10.7717/peerj.11359_

## Round 0.1 · original submission · Minor Revisions

Dear Author, your manuscript has been extensively reviewed and the comments you can find attached. Both reviewers suggested minor revision and I side with them on this matter.

Reviewer 1 ·

Basic reporting

This is a well-written, well-organized and timely manuscript. The manuscript is remarkably clear overall and conforms to all relevant professional standards. The introduction is well-written, but focusing more clearly and directly on the most interesting question would enhance the manuscript. Certainly, exploration of the maize microbiome among different varieties and plant locations is interesting, but the manuscript contains a somewhat “buried headline” – the fact that the majority of isolates they studied were not able to be accurately identified by the rather ubiquitous biotyping workflow is very important and likely of great interest to the diverse readers of PeerJ. The need for methods to enhance rapid MALDI-based approaches to explore non-clinical realms of the microbiome are very much needed, and this paper directly addresses that need; however, this research question/need could be more fully and clearly foregrounded and underscore the coherent nature of this part of the body of work. Figures are clear, and raw data are available. A table summarizing the number of isolates analyzed, the number of isolates per library, etc. would be a helpful addition.

Experimental design

The work is certainly within the scope of PeerJ, and the research questions are intriguing and relevant; however, they should be better focused as detailed above (more clearly focusing on and underscoring the need for rapid MALDI-based approaches to environmental microbiome analysis). The analysis appears appropriately rigorous and quantitative largely, but some details are not included (or difficult to locate), including numbers of replicates per sample/library, number of laser shots, raster spots, etc.

Validity of the findings

The findings seem valid and well-substantiated by the data presented, but additional background/rationale is needed in some areas (see comments in marked up manuscript attached). Combining the results and discussion sections may facilitate the authors’ ability to better contextualize the findings.

Additional comments

Overall, this is a timely manuscript and notably well-written and clear. The concept of MTUs is rather intriguing (and somewhat novel), but the concept warrants greater focus (again, foregrounding) and clear explanation (early in the manuscript) its significance and all methods used to define an MTU. The discussion section has a nicely concise sentence or two summarizing how this is calculated/used, but this ought to be included earlier in the manuscript (perhaps even discussing the need for something akin to an MTU in the Introduction) and more clearly explained in less jargon-laden prose for the benefit of the diverse readers of PeerJ.

Annotated reviews are not available for download in order to protect the identity of reviewers who chose to remain anonymous.

Reviewer 2 ·

Basic reporting

In the introduction, there is little or no time spent on the benefits of increased resolution in phenotyping bacteria, which is a big limitation of 16S identification (which lumps lots of functionally distinct bacteria into the same taxonomic bin).

In the discussion, I don’t think you need to spend several paragraphs listing all the different types of bacteria that you observed (unless you can speculate at the level of bacterial phenotype on why a genus like Pseudomonas was grouped into different MTUs), rather try to explain how this method could yield phenotypic data about the microbes (eg. a difference in spectra may reflect production of growth promoting hormones by beneficial vs neutral strains) or talk about the advantages and potential of expanding it to help study other plant associated microbes. Please discuss whether Maldi-TOF technology is commercially available at reasonable prices to most researchers.

I have made various other comments or suggestions on the attached PDF, which I hope you'll find helpful.

Experimental design

no comment

Validity of the findings

no comment

Additional comments

Your study is an important one, and you are beginning to build resources that the plant-microbiome community needs, thank you. The scientific importance of this paper, is its development of a method (and a database?) for better identification and characterization of plant-associated bacteria, however the authors present it more as a study of maize microbial ecology. Beginning with the title, I think your paper could benefit from a restructuring of the orientation, for example remaking it as something like this: “Development of an inexpensive matrix-assisted laser desorption – time of flight mass spectrometry method and database for the identification of rhizobial and endophytic bacteria”. Maize microbes were really just a model system, and shouldn't be built up into something too much more. I think that you should also try to better introduce and discuss more about what the differences in the metabolic spectra might mean and how it might be developed in the future to help better characterize the phenotypes of plant-associated bacteria. For example, this paper (Clark, Chase M., et al. "Coupling MALDI-TOF mass spectrometry protein and specialized metabolite analyses to rapidly discriminate bacterial function." Proceedings of the National Academy of Sciences 115.19 (2018): 4981-4986.) reported examples of how changes in a bacteria's functional proteins alter the MALDI-TOF profile. I think you should likewise add and restructure the introduction and discussion to cover some of the points I mention in the "Basic Reporting" area. I have made various other comments or suggestions on the attached PDF, which I hope you'll find helpful.

Annotated reviews are not available for download in order to protect the identity of reviewers who chose to remain anonymous.

---

## Round 0.2 · accepted · Accept

Dear Author, I am happy to inform you that your manuscript has been accepted for publication.